# Dynamics of individual molecular shuttles under mechanical force

Teresa Naranjo[1], Kateryna M. Lemishko[2], Sara de Lorenzo[1], Álvaro Somoza[1,2], Felix Ritort[3,4], Emilio M. Pérez [1] & Borja Ibarra [1,2]

Molecular shuttles are the basis of some of the most advanced synthetic molecular machines. In these devices a macrocycle threaded onto a linear component shuttles between different portions of the thread in response to external stimuli. Here, we use optical tweezers to measure the mechanics and dynamics of individual molecular shuttles in aqueous conditions. Using DNA as a handle and as a single molecule reporter, we measure thousands of individual shuttling events and determine the force-dependent kinetic rates of the macrocycle motion and the main parameters governing the energy landscape of the system. Our findings could open avenues for the real-time characterization of synthetic devices at the single molecule level, and provide crucial information for designing molecular machinery able to operate under physiological conditions.

[1] IMDEA Nanociencia, C/Faraday 9, Ciudad Universitaria de Cantoblanco, 28049 Madrid, Spain. [2] Nanobiotecnología (IMDEA-Nanociencia) Unidad Asociada al Centro Nacional de Biotecnología (CSIC), 28049 Madrid, Spain. [3] Condensed Matter Physics Department, C/Marti i Franques s/n, University of Barcelona, 08028 Barcelona, Spain. [4] CIBER-BBN de Bioingeniería, Biomateriales y Nanomedicina, Instituto de Sanidad Carlos III, Madrid, Spain. These authors contributed equally: Teresa Naranjo and Kateryna M. Lemishko. Correspondence and requests for materials should be addressed to E.M.P. (email: emilio.perez@imdea.org) or to B.I. (email: borja.ibarra@imdea.org)

Recent advances in synthetic molecular machinery have led to systems with the ability to work as molecular motors[1,2]. Many of these molecular devices are currently the subject of intense research interest due to their many potential applications in diverse fields from biomedicine to nanorobotics and their kinetics and thermodynamics in bulk are well understood[3]. However, quantitative information on the dynamics and mechanistic principles behind the operation of individual synthetic molecular devices is still lacking[4]. Single-molecule force spectroscopy techniques, such as atomic force microscopy (AFM), optical and magnetic tweezers, have revolutionized our understanding of the dynamics of biological motors[5,6], but they have not yet had impact on the field of synthetic molecular machines. To date, AFM has been the most widely used technique to pull on synthetic mechanically interlocked molecules in order to investigate their mechanochemical properties[7–12]. In recent years, optical tweezers have revealed themselves as a versatile, powerful single-molecule manipulation technique to study biology at the single-molecule level. By attaching the system under study to an external probe (a microscopic bead acted upon by a photonic field) it is possible to measure and control in real-time the nanometer displacements (or conformational changes) and picoNewton forces (<10 pN) that define the operational dynamics of biological systems under aqueous conditions[5,13,14].

Here, we exploit the properties of optical tweezers to quantify the real-time operation of individual synthetic molecular shuttles. In molecular shuttles, a macrocycle trapped onto a linear component (axle) can be moved reversibly between two or more portions of the axle (stations), in response to external stimuli[15–17]. These architectures are the basis for some of the most advanced examples of synthetic molecular machines[18–22]. We show that coupling of a molecular shuttle with double-stranded DNA (dsDNA) enables its single-molecule manipulation under aqueous conditions. This strategy enables us to determine key parameters of the operation of this synthetic device under near-physiological conditions, including information that cannot be obtained from bulk approaches. First, we measure the mechanical strength of the macrocycle at each station and the free energy of the shuttling reaction. Second, we quantify the force-dependent real-time shuttling kinetics of the macrocycle between the two stations and calculate the energy landscape of the system under our experimental conditions.

## Results

### Chemical design and experimental setup

We synthetized a hydrogen-bonded Leigh-type molecular shuttle, featuring a tetraamide macrocycle locked onto a oligoethyleneglycol axle by two diphenylethyl groups (stoppers) situated at either end of the axle, Fig. 1a. The axle features fumaramide (*fum*) and succinic amide-ester (*succ*) stations, each of which can establish up to four hydrogen bonds with the macrocycle with sufficiently different affinities to yield a *fum:succ* occupancy ratio biased towards the *fum* station, even in the strongly competing solvent $d_6$-DMSO (Supplementary Methods). To interface the synthetic device with the optical tweezers, a single shuttle was connected between two functionalized beads using two dsDNA molecules, Fig. 1b. The first dsDNA connects the macrocycle with the bead in the optical trap and serves as a handle to control and trace the stochastic motion of the macrocycle between the two stations. The second dsDNA connects the stopper adjacent to the *fum* station to a bead on top of a micropipette, providing physical separation (~282 nm) between them. Experiments were performed under aqueous conditions (20 mM Tris-HCl pH 7.5, 150 mM NaCl) at 22 ± 1 °C.

### Determination of the mechanical strength of the macrocycle at each station

In order to investigate the mechanical strength of the interaction of the macrocycle at each station, individual rotaxane-DNA hybrids were subjected to pulling-relaxing cycles by moving the optical trap relative to the micropipette at a fixed pulling rate of 200 nm s⁻¹ (Fig. 2a). At low pulling forces, the force-extension curves fit well to a worm-like chain model (WLC) of a polymer with a persistence length of 50 nm, characteristic of a single dsDNA molecule[23]. At these forces, the macrocycle resides over the thermodynamically preferred *fum* station. Once the force exceeds the strength of the noncovalent interactions holding the macrocycle at the *fum* station, an abrupt increase in extension is observed. Using the WLC model, we calculate the additional contour length released during this transition, $\Delta L_c$. The average $\Delta L_c$ was 15 ± 2 nm, which is consistent with the calculated distance between stations in the fully extended shuttle (11.2 nm, Supplementary Fig. 1). This is an unambiguous confirmation that the peak corresponds to the force-induced shuttling of the macrocycle from *fum* to *succ* stations. When relaxing the force, a sudden extension decrease to the original position is observed, that reflects the ability of the macrocycle to exert a force against the load to return to the thermodynamically favored *fum* station[8]. The robustness of our system allows the collection of up to one hundred force-extension curves of each rotaxane-DNA hybrid (Supplementary Fig. 2) and the construction of detailed histograms for the rupture forces at each station (Fig. 2b). The average rupture forces are $f_{fum} = 8.8 \pm 0.6$ pN and $f_{succ} = 8.1 \pm 0.5$ pN (Fig. 2b), which are comparable with those required to break an equivalent number of H-bonds in biological and synthetic systems under similar conditions[24,25]. In addition, the intersection of the force distributions denotes the coexistence force, $f_{1/2}$, or the force under which the macrocycle has an equal probability of residing over the *fum* or *succ* stations, $f_{1/2} = 8.46$ pN. Assuming that the average dissipated work in our system behaves linearly with the pulling speed, as showed for other bi-stable systems, like RNA hairpins[26], the total free energy of shuttling can be calculated as the product of the coexistence force ($f_{1/2}$) and $\Delta L_c$, giving $\Delta G = 31 \pm 2$ $k_B T$ (18 ± 1 kcal mol⁻¹)[27,28]. $\Delta G$ equals the free energy of shuttling from the *fum* to the *succ* station at zero force plus the free energy for stretching the rotaxane-DNA hybrid from a force of 0 pN to $f_{1/2}$, which we measured as 11.8 $k_B T$ (7.1 kcal mol⁻¹, Supplementary Fig. 3)[29]. Therefore, the free energy of shuttling of the macrocycle from the *fum* to the *succ* station at zero force under near-physiological conditions is 19 ± 2 $k_B T$ (11 ± 1 kcal mol⁻¹).

### Determination of the force-dependent shuttling rates

In order to investigate the dynamics and bi-stability of the system we imposed a constant force around $f_{1/2}$ with a feed-back stabilized protocol capable of maintaining a preset force within ~0.05 pN. Under these conditions, hundreds of shuttling events of the macrocycle between the two stations were monitored in real-time during minutes (Fig. 3a and Supplementary Fig. 4). These shuttling events present well-defined residence times at each station, in which the macrocycle undergoes thermal motion near one station, and much faster transition times from one station to the other, a finding consistent with models[30] (Fig. 3b). The residence times were exponentially distributed, as expected for a two-state system in equilibrium (Supplementary Fig. 5)[31]. At forces ~ $f_{1/2}$ (Fig. 3, middle column), the macrocycle spends roughly equal amounts of time at each station. Under these conditions, the histogram of extensions shows two similar peaks, which fit well to two Gaussian curves separated by an average $\Delta x = 15.5 \pm 2.5$ nm (Fig. 3c), identical to the contour length increase

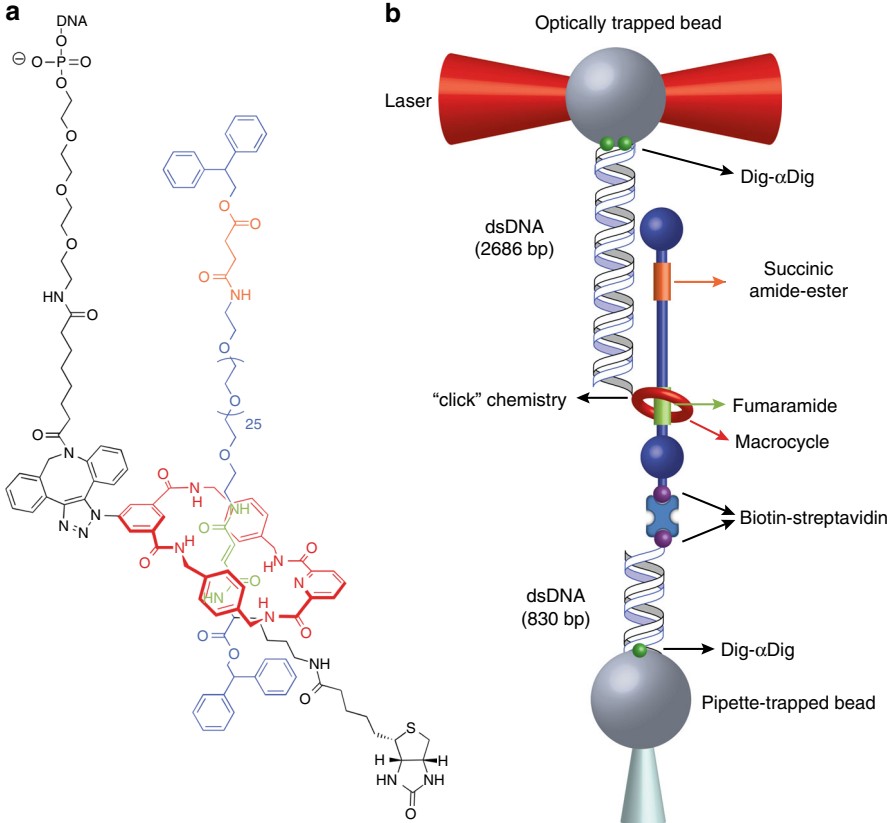

**Fig. 1** Molecular design and experimental setup. **a** The chemical structure of the molecular shuttle consists of a benzylic tetraamide macrocycle (red) locked onto an axle (blue) terminated by diphenylethyl groups at each end (stoppers). The axle contains fumaramide (*fum*, green) and succinic amide-ester (*succ*, orange) stations separated by a long oligoethyleneglycol spacer. At low forces the macrocycle resides predominantly over the *fum* station. A DNA oligonucleotide was covalently attached to the macrocycle to allow for manipulation in the optical tweezers. **b** The molecular shuttle is attached to two polystyrene beads (diameter ~3 μm) through two dsDNA molecules (not to scale): The 2686 bp dsDNA molecule connects the macrocycle to the bead in the optical trap ($k = 0.13$ pN nm$^{-1}$, laser power = 60 mW) via digoxigenin-antidigoxigenin (Dig-αDig) connections. The 830 bp dsDNA molecule connects the shuttle to a bead held by suction on a micropipette, via biotin-streptavidin connections at one end and Dig-αDig at the other end. Load is applied to the system by moving the optical trap away from the micropipette. At constant load, the shuttling dynamics of the macrocycle are inferred by measuring the motions of the bead in the optical trap

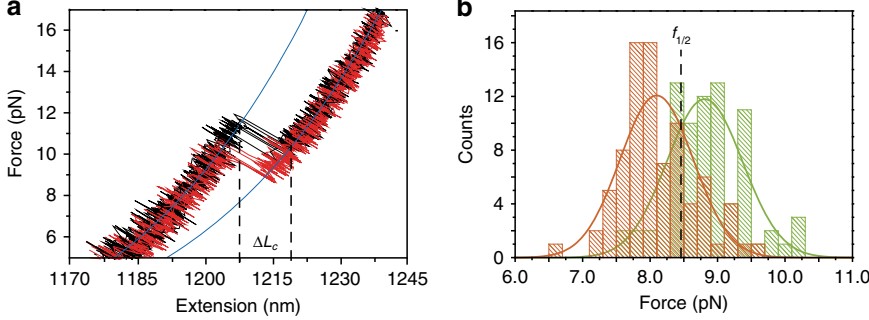

**Fig. 2** Determination of breaking forces at each station. **a** Representative pulling (black) and relaxing (red) curves ($N = 6$) recorded at a pulling rate of 200 nm s$^{-1}$ displaying WLC behavior of dsDNA handles (blue line). Please, see Methods section for calculation of end-to-end distances. WLC fit to the contour length increase after the shuttling event gives $\Delta Lc$ ~15 nm. **b** The intersection of the distributions of the breaking forces at *fum* (green Gaussian fit) and *succ* (orange Gaussian fit) stations reveals the coexistence force, $f_{1/2} = 8.46$ pN ($N = 450$ curves)

after the shuttling event measured in the pulling-relaxing experiments. By gradually varying the force on the system, it was possible to displace the equilibrium favoring occupancy of either the *fum* or *succ* bound states (Fig. 3 left and right columns). From the extension histograms at different forces, we obtain directly the energy profiles, using Boltzmann's distribution

(Fig. 3d). Within the force range studied, the profiles show a well-defined transition state between two broad minima, the relative height of which varies with force.

**Force-dependent kinetic rates of the shuttling process.** To investigate the force dependence of the shuttling rates (inverse of

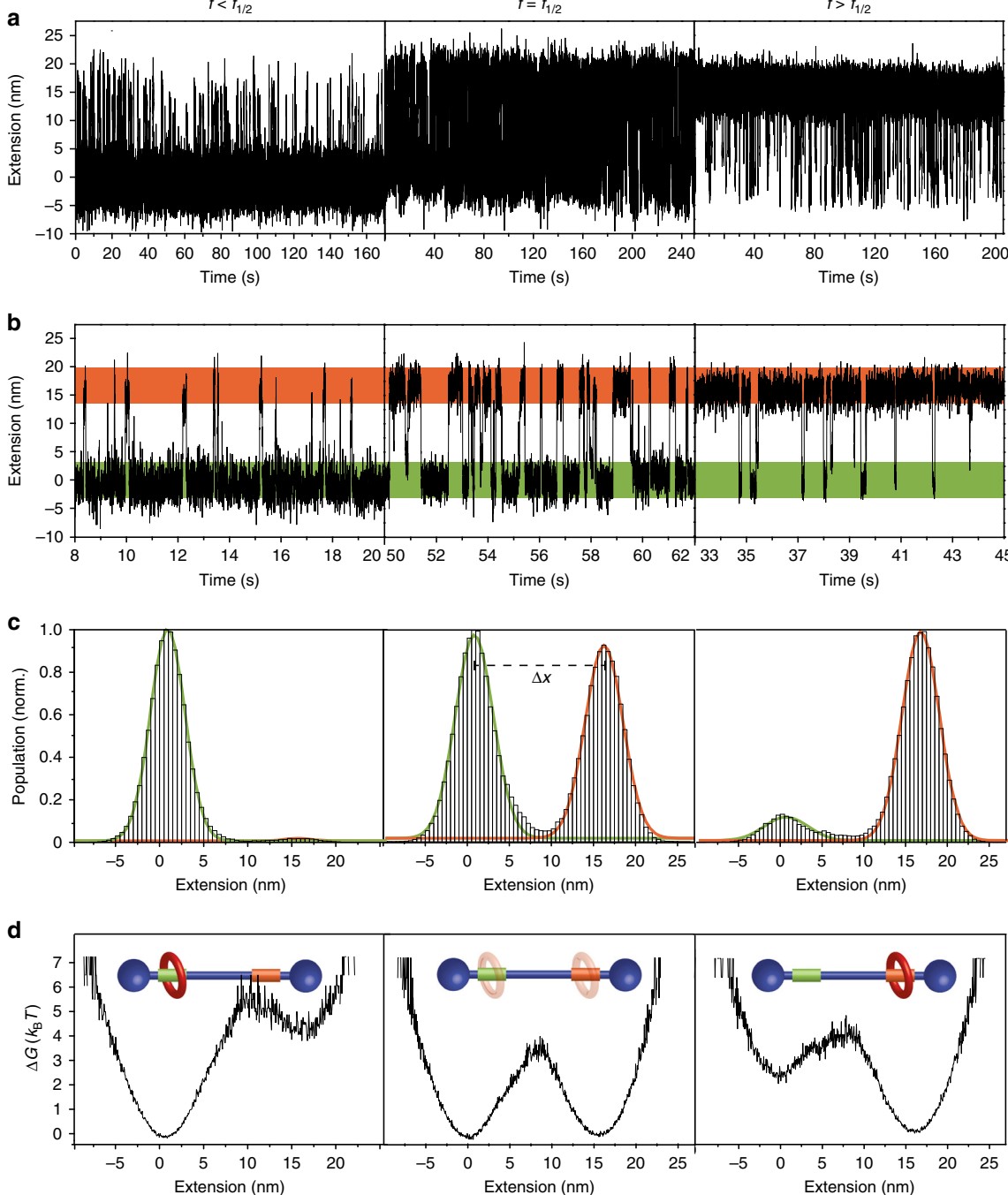

**Fig. 3** Shuttling dynamics, positional control and energy profiles. **a** Extension vs. time data for single molecular shuttles at different constant forces. **b** Zoom-in on the data presented in a at arbitrary time intervals, showing residence and transition times. An average of 400 shuttling events were recorded for each molecular shuttle ($N = 24$). **c** Extension histograms, fitted with two Gaussian curves (orange and green lines) affording a shuttling distance of $\Delta x$ ~ 15 nm. **d** Energy profiles at each force, obtained directly from the relative populations depicted in (**c**). At force < $f_{1/2}$ (8.1 pN, left column), the macrocycle occupies preferentially the *fum* station (green). At force ~ $f_{1/2}$ (8.5 pN, middle column), both stations present similar occupancies . At force > $f_{1/2}$ (9.6 pN, right column), the macrocycle occupies preferentially the *succ* station (orange)

residence times) from *fum* to *succ* station, $k_{fum}$, and its reverse $k_{succ}$, we measured the rates of shuttling of 24 different molecular shuttles at several constant forces around $f_{1/2}$ (Fig. 4). The experimental values of $k_{fum}$ and $k_{succ}$ present an exponential dependence on force and fit well to the Bell-Evans model[32–35] along the entire range of forces measured, confirming again that the shuttling occurs under near equilibrium conditions (Fig. 4). From the fits, we can extract the main parameters that

characterize the free energy landscape of the shuttling process (Methods). The intersection of the two linear fits rendered the coexistence force, $f_{1/2} = 8.83$ pN, which is consistent with the $f_{1/2}$ value obtained from the rupture force distributions in the pulling-relaxing experiments. This value, together with the average shuttling distance measured between the two stations in these experiments, $\Delta x = 15.5 \pm 2.5$ nm, yield $\Delta G = 33 \pm 2$ $k_B T$ ($20 \pm 1$ kcal mol$^{-1}$, where $\Delta G = f_{1/2} \cdot \Delta x$). These experiments yield a direct

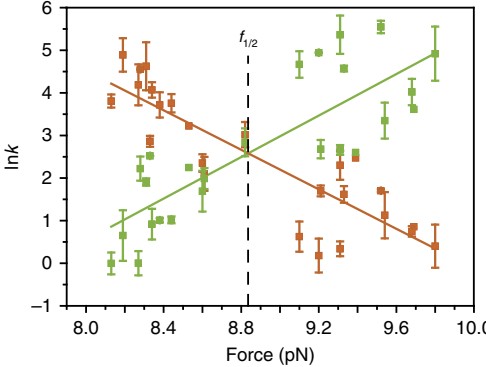

**Fig. 4** Dependence of shuttling rates with force. The shuttling rates from *fum* to *succ* station, $k_{fum}$ (green), and the backward rates, $k_{succ}$ (orange), vary exponentially with force. The values of ln$k$ are represented as mean ± standard deviation. Fits of the data with the Bell-Evans theory revealed the coexistence force, $f_{1/2}$, and the main parameters of the energy landscape

calculation of $\Delta G$ based on the population distributions, and agree well with the value for $\Delta G$ obtained with the pulling-relaxing experiments described above. In addition, the results of the fits rendered the relative distances from the *fum* and *succ* stations to the transition state at $f_{1/2}$, $x_{fum} = 10 \pm 2$ nm and $x_{succ} = 6 \pm 2$ nm, respectively. Therefore, for these particular experimental conditions, the distance to the transition state at the coexistence force is asymmetric: the macrocycle needs to travel a longer distance to reach the point where the probability of completing the shuttling process is 50% after abandoning the stronger *fum* station, compared to the weaker *succ* station.

In summary, our experimental design allowed the measurement of hundreds of pulling-relaxing cycles and real-time shuttling events on tens of individual molecular shuttles. From these data, we obtain crucial information that is not accessible from bulk experiments, such as the mechanical stability of the macrocycle at each station, the dynamics of shuttling, and a clear picture of the energy landscape at different forces, which reveals broad energy minima separated by a well-defined transition state, positioned asymmetrically with respect to each station. This experimental picture is significantly different from idealized theoretical energy profiles[3]. This experimental system may enable the study of the combined effect of force and other variables (temperature, light, solvent composition) on the operation of individual molecular shuttles under aqueous conditions. The results from these studies facilitate more reliable comparisons with biological devices (also operating in water) and, therefore, help to design more efficient synthetic and hybrid molecular devices. From a more general point of view, once the synthetic and technological challenges to study the thermodynamics and real-time kinetics of individual synthetic molecular devices have been surmounted, the atomic precision of synthetic chemistry will allow addressing fundamental questions, such as the structural basis for microscopic reversibility or the controversial presence (or absence) of quantum effects in nanoscopic engines[36].

## Methods
**Molecular shuttle synthesis**. All solvents were dried according to standard procedures. Reagents were used as purchased. All air-sensitive reactions were carried out under argon atmosphere. Flash chromatography was performed using silica gel (Merck, Kieselgel 60, 230–240 mesh, or Scharlau 60, 230–240 mesh). Analytical thin layer chromatographies (TLC) were performed using aluminum-coated Merck Kieselgel 60 F254 plates. NMR spectra were recorded on a BrukerAvance 400 ($^1$H: 400 MHz; $^{13}$C: 100 MHz; COSY: 400 MHz; HSQC: 400 MHz), spectrometer at 298 K, unless otherwise stated, using partially deuterated solvents as internal standards. Coupling constants ($J$) are denoted in Hz and chemical shifts ($\delta$) in ppm.

Multiplicities are denoted as follows: s = singlet, d = doublet, t = triplet, m = multiplet, br = broad. Fast Atom Bombardment (FAB) and Matrix-assisted Laser desorption ionization (coupled to a Time-Of-Flight analyzer) experiments (MALDI-TOF) were recorded on a VS AutoSpec spectrometer and a Bruker ULTRAFLEX III spectrometer, respectively. A detailed description of the synthesis of the molecular shuttle is provided in Supplementary Methods.

**Oligonucleotide synthesis**. The syntheses of oligonucleotides were performed on a MerMade 4 synthesizer (BioAutomation Corporation). For each oligonucleotide synthesis, columns were filled with the corresponding Controlled Pore Glass (CPG) solid support. Anhydrous MeCN was used as solvent. For the cleavage of DMTr protecting groups, the resin was purged with 3% trichloroacetic acid in anhydrous DCM. The removal of the acid was carried out by purging with anhydrous MeCN. The activation of the phosphoramidite functionality was effected by a 0.25 M benzylthiotetrazole solution in anhydrous MeCN. The coupling time for standard phosphoramidites was 2 min and for cyclooctine derivative 5 min. Oxidation of P (III)-species was attained by alkaline iodine solution (20 mM I$_2$ in THF/Py/water 7/2/1). For the capping of residual 5′-OH-groups, a mixture of solution A (10% Ac$_2$O, 10% pyridine, 80% THF) and solution B (10% 1-methylimidazole in THF) was used. After completion of the synthesis, the oligonucleotides were cleaved from the solid support with concomitant removal of the Fmoc and β-cyanoethyl protecting groups by reacting the oligonucleotide-charged solid support with 28% aq. NH$_3$ at 55 °C for 20 h. The solution was filtered and the filtrate was concentrated in vacuum. The residue was dissolved in 750 μL water. For purification of this crude oligonucleotide solution, a volume containing ~40 nmol crude oligonucleotide was applied to gel electrophoresis (1 mm, 20% polyacrylamide). The oligonucleotide-containing segments of the gel were visualized by UV-light (260 nm) and separated from the rest of the gel. Oligonucleotides (R- 5′TTTTTTTTTTTTTTTTTAGCT; 3′ AAAAAAAAAAAAAAAA5′) were extracted from the gel using an elutrap system (3 h, 200 V). The solutions were desalted using a NAP-10 column and concentrated in an evaporating centrifuge.

**Copper-free "click" reaction for coupling the macrocycle with a ssDNA oligonucleotide**. To a stirred solution of oligonucleotide 1 (50 μM) in H$_2$O/DMF (0.3 mL, 4/1) was added rotaxane 15 (60 μM) in H$_2$O/DMF (0.3 mL, 4/1). The mixture was allowed to stir overnight at room temperature. Followed and purified by gel electrophoresis (0.2 mm, 20% polyacrylamide). The oligonucleotide-containing segments of the gel were visualized by UV-light (260 nm) and separated from the rest of the gel. Oligonucleotides were extracted from the gel using an elutrap system (3 h, 200 V). The solutions were desalted using a NAP-10 column and concentrated in an evaporating centrifuge, to give the oligonucleotide 3 (Supplementary Fig. 6).

**Length of the oligoethyleneglycol axle**. The theoretical length between the *fum* and *succ* stations of the relaxed axle in the gas phase was calculated as 11.2 nm by geometry optimization with the molecular mechanics (MMF94s, Supplementary Fig. 1) by using Avogadro[37].

**Coupling of the shuttle with DNA handles**. For optical tweezers experiments a single shuttle was connected between two functionalized beads using two dsDNA spacers, Fig. 1b. The first dsDNA spacer (or dsDNA handle) connects the macrocycle with the bead in the optical trap. This dsDNA handle was attached to the macrocycle as follows: briefly, we first obtained a rotaxane modified with a ssDNA oligonucleotide through a 1,2,3-triazole group in the macrocycle (Supplementary Methods). Then the oligonucleotide on the rotaxane was annealed with a complementary polydA oligonucleotide (oligo 2). Finally, a 2686 bp long dsDNA molecule containing a protruding TCGA 5′-end was ligated to the protruding AGCT 3′-end of the oligo-rotaxane construct. This dsDNA molecule was labeled with multiple digoxigenins (Dig) at the other end following standardized procedures[38]. A shorter dsDNA molecule (830 bp), labeled with streptavidin in one end and Dig in the other, connects the biotin labeled stopper to an anti-Dig covered bead held by suction on a micropipette.

**Synthesis of DNA spacers**. The 2686 dsDNA handle was derived from the puc19 vector (Novagen). The plasmid was cut with *BamHI* and *SacI* restriction endonucleases (NEB). *BamHI* end was labeled with multiple Dig as described before. The 830 bp dsDNA spacer connecting the biotin labeled stopper to the bead on the micropipette was synthetized by PCR amplification of the poly-linker section of the puc19 vector with two DNA oligonucleotides: one labeled with biotin and the other with digoxigenin at the 5′-end. The spacer was then incubated with saturated amounts of streptavidin and attached to anti-digoxigenin covered polystyrene micron-beads (Spherotech, Co).

**Optical tweezers experiments**. Optical tweezers experiments were performed with a highly stable miniaturized dual-beam optical tweezers[39], with a limiting spatial resolution of ≤ 1 nm. In the optical tweezers the molecular shuttle is attached between two beads thought two dsDNA molecules as described above and

in the main text (Fig. 1). All force-extension curves were obtained at a pulling rate of 200 nm s$^{-1}$. The number of cycles obtained per molecule range between 20 and 120. Our optical tweezers setup controls the distance between the trap center and the pipette ($X_{total}$), which is different from the end-to-end distance of the molecular system we are interested in ($X_{end-end}$). The end-to-end distance of the system is calculated as follows:

$$X_{end-end} = X_{Tot} - F/k$$

where $F$ is the force applied to the system, $k$ the stiffness of the trap, and $F/k$ corresponds to the distance (nm) moved by the bead out of the trap center (bead position). We assume a linear spring restoring force. The stiffness of the trap depends on the power of the lasers, the size and the index of refraction of the bead on the trap and the viscosity of the surrounding medium. The value of the stiffness obtained in our experimental conditions is $k = 0.135 \pm 0.0043$ pN nm$^{-1}$ (for 3 μm beads with a laser power of 60 mW).

Constant force measurements were performed with a feed-back stabilized protocol capable of maintaining a preset force within ~0.05 pN by moving the beads closer or further apart. All experiments were performed under near-physiological conditions (20 mM Tris-HCl pH 7.5, 150 mM NaCl) at 22 ± 1 °C and data was taken at 1 kHz.

**Residence time calculations**. In the hopping experiments the residence times at each station were calculated by partitioning each trace into two states using a threshold set at the midpoint between the histogram peaks.

**Fit to Bell-Evans theory**. The shuttling rates from *fum* to *succ* station, $k_{fum}$, and its reverse $k_{succ}$, were found to depend exponentially on force (Fig. 4). According to the Bell-Evans model $k_{fum}$, and $k_{succ}$ depend on force following:

$$k_{fum}(f) = k(0)\exp\left(\frac{f\,x_{fum}}{k_B T}\right)$$

$$k_{succ}(f) = k(0)\exp\left(\frac{\Delta G - f\,x_{succ}}{k_B T}\right)$$

where $f$ corresponds to the external force, $k(0)$ is the shuttling rate at zero force; $x_{fum}$ and $x_{succ}$ are the relative distances between the *fum* and *succ* stations to the transition state at $f_{1/2}$, respectively, and $\Delta G$ is the shuttling free-energy of the molecular switch at zero force, defined as $\Delta G = f_{1/2} \cdot \Delta x$ (where $\Delta x$ is the distance between the *fum* and *succ* stations).

**Calculation of energy profiles**. The shuttling potential energy profile at a given force can be computed according to Boltzmann distribution as follows: $E(x) = -k_B T \cdot \ln\rho(x)$. The term $\rho(x)$ corresponds to the probability distribution of an extension ($x$) and was obtained from the extension distributions shown in Fig. 3c.

## Data availability
All experimental data are available upon request.

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

## Acknowledgements

We thank J. M. R. Parrondo, P. J. de Pablo, I. A. Rodriguez and E. R. Kay for comments on preliminary versions of this article, and members of the E.M. Pérez and B. Ibarra labs for useful discussions. This work was supported by the European Research Council (ERC-StG-MINT 307609), the Ministerio de Economía y Competitividad (grants BFU2015-63714-R, CTQ2014-60541-P, CTQ2017-86060-P, SAF2014-56763-R, FIS2016-80458- P). IMDEA Nanociencia acknowledges support from the "Severo Ochoa" Programme for Centers of Excellence in R&D (MINECO, Grant SEV-2016-0686). K.M.L was supported by the Ministerio de Educación Cultura y Deporte (FPU2014/06867). F.R. acknowledges support from ICREA Academia 2013.

## Author contributions

T.N. performed the chemical synthesis and K.M.L. the single-molecule experiments. T.N. and K.M.L. analyzed the data. F.R. led data analysis and interpretation. A.S. and S.d.L. supplied analytical and technical tools. E.M.P and B.I. conceived, designed, and directed the study and wrote the paper, with contributions from all authors.

## Additional information

**Competing interests:** The authors declare no competing interests.

