## [Peer Review File · Nature Communications]

This manuscript has been previously reviewed at another journal that is not operating a transparent peer review scheme. This document only contains reviewer comments and rebuttal letters for versions considered at Nature Communications.

Reviewers' Comments:

Reviewer #1:

Remarks to the Author:

This manuscript is very well written; however it is very similar to a significant body of work involving the force spectroscopy of the unfolding of single biomolecules. But it does differ: Instead of probing the disruption of a series of interactions within a fully-covalent structure, this work probes the displacement of an interlocked macrocycle of a 2-state rotaxane, to estimate ΔG or the minimum work required to switch to the unfavourable state of a particular 2-state rotaxane. There are other force measurements of 2-state rotaxanes and the authors do cite these. This is a very clean piece of work following previously published procedures and the authors should be congratulated on this accomplishment. However, the new results are specific to the 2-state rotaxane and an argument could be made that this is more appropriate in a more targeted chemistry journal. I have only two comments to add.

The authors state that this is a near equilibrium measurement of ΔG of a 2-state rotaxane - and the reasoning from about line 80 focusses upon the statistics of forces measured (Fig 2 B) as per Crooks fluctuation theorem. However the measurement of the transition from fums to sec state is clearly not near-equilibrium on the timescale of the experiment - Fig 2 A shows that there is no measurements taken as the ring traverses between the states - the transition between states occurs on a faster timescale than the experiment. Consequently, this is not a near-equilibrium measurement of properties of that two state transition. Admittedly, several other papers in the biological force measurement field make similar statements. What is at near-equilibrium is the DNA handles - the force measurement is tuned to measuring the stretching of DNA, and not to the transition between states in the interlocked molecule.

Finally, the last sentence is the authors general point of view is confusing. The authors refer to the challenges of studying the thermodynamics and real time kinetics - and their work nicely shows how previously applied techniques in biological field can be cleanly applied to these 2-state rotaxanes. However they state that once challenges are surmounted, then this will allow scientists to investigate the "controversial presence (or absence) of quantum effects in nanoscopic engines" - however, the quantum effects are evident at different time, energy, and length scales - and this is not discussed in the paper and this manuscript does not in any way suggest how to do that. So I can accept this as a generic statement - but what is worrying is that even within this optical tweezers-based force measurement, the time scales of this fast 2-state molecular transition and the timescale of the force measurement are not correctly described (its not close to equilibrium).

Point-by-point response:

Dynamics of individual molecular shuttles under mechanical force

NCOMMS-18-23031-T

Reviewer#1:

This manuscript is very well written; however it is very similar to a significant body of work involving the force spectroscopy of the unfolding of single biomolecules. But it does differ: Instead of probing the disruption of a series of interactions within a fully-covalent structure, this work probes the displacement of an interlocked macrocycle of a 2-state rotaxane, to estimate ΔG or the minimum work required to switch to the unfavourable state of a particular 2-state rotaxane. There are other force measurements of 2-state rotaxanes and the authors do cite these. This is a very clean piece of work following previously published procedures and the authors should be congratulated on this accomplishment. However, the new results are specific to the 2-state rotaxane and an argument could be made that this is more appropriate in a more targetted chemistry journal. I have only two comments to add.

The authors state that this is a near equilibrium measurement of ΔG of a 2-state rotaxane - and the reasoning from about line 80 focusses upon the statistics of forces measured (Fig 2 B) as per Crooks fluctuation theorem. However the measurement of the transition from fums to sec state is clearly not near-equilibrium on the timescale of the experiment - Fig 2 A shows that there is no measurements taken as the ring traverses between the states - the transition between states occurs on a faster timescale than the experiment. Consequently, this is not a near-equilibrium measurement of properties of that two state transition. Admittedly, several other papers in the biological force measurement field make similar statements. What is at near-equilibrium is the DNA handles - the force measurement is tuned to measuring the stretching of DNA, and not to the transition between states in the interlocked molecule.

We thank Rev#1 for raising this point, which we may not have explained clearly in the text. With the expression *near equilibrium*, we intended to say that our pulling experiments take place within the linear response regime. This means that the average dissipated work (equal to the difference between the average total work and the reversible work) behaves linearly with the pulling speed, which is the situation encountered in our experiments (see for example, Manosas et al, J. Stat. Mech (Theor. and Exp) (2009) P02061 for a discussion on this). To avoid possible confusions by using the term "near equilibrium", we have decided to replace it by the explanation provide above (including the new reference).

Finally, the last sentence is the authors general point of view is confusing. The authors refer to the challenges of studying the thermodynamics and real time kinetics - and their work nicely shows how previously applied techniques in biological field can be

cleanly applied to these 2-state rotaxanes. However they state that once challenges are surmounted, then this will allow scientists to investigate the "controversial presence (or absence) of quantum effects in nanoscopic engines" - however, the quantum effects are evident at different time, energy, and length scales - and this is not discussed in the paper and this manuscript does not in any way suggest how to do that. So I can accept this as a generic statement - but what is worrying is that even within this optical tweezers-based force measurement, the time scales of this fast 2-state molecular transition and the timescale of the force measurement are not correctly described (its not close to equilibrium).

As Rev#1 points out, the last phrase of the manuscript is a generic statement. We are confident that future technological advances in the single-molecule detection and manipulation field, will provide access to the temporal resolution required to measure in much greater detail the inner workings of synthetic molecular devices.